# Is Life Unlivable for Youth in Post-DEI America?: Understanding Rising Suicide Rates Across Diverse Youth Groups Through Traditional Suicide Paradigms

**DOI:** 10.3390/healthcare13202585

**Published:** 2025-10-14

**Authors:** Mimi Yen Li, Christina Mata, Kalpana Nathan

**Affiliations:** Department of Psychiatry and Behavioral Sciences, University of California, San Francisco, CA 94107, USA; christina.mata@ucsf.edu (C.M.);

**Keywords:** suicide, adolescent, DEI, livability, LGBTQ+, youth of color, incels

## Abstract

We pose the question of whether life has become unlivable for the young in America amidst the current political climate, which has systematically deregulated our social structures that safeguard against oppressive and unjust practices. What leads the young to become demoralized to the point of wanting to end their lives? Drawing on several established psychosocial models for suicide, including those of Durkheim, Joiner, and Butler, we highlight how groups of youth as disparate as youth of color, LGBTQ+ youth, and young men experience unique sociopolitical stressors that contribute to increased suicidality. We argue that despite differences in their contexts, they experience shared pathways to suicide. At a time when U.S. funding cuts threaten to dismantle the progress made in recent years to address structural racism and sexism, we also make a case for the importance of mental health clinicians’ engagement in advocacy work that recognizes the sociopolitical influences on mental health and highlight universal school-based social emotional learning (USB SEL) as one beneficial intervention to target mental health outcomes across disparate youth groups.

## 1. Introduction

Youth suicide is a major public health concern globally and is the third leading cause of death among 15 to 29-year-olds [1]. In the United States, suicide is the second leading cause of death among 10 to 19-year-olds, and one in five youth have reported seriously contemplating suicide [2,3]. Suicide deaths among 10 to 24-year-olds in America increased by 62% from 2007 to 2021; they have also risen dramatically in American preteens as young as 8 years old, with an 8.2% annual increase from 2008 to 2022 [4]. High suicide rates are found in subsets of youth as diverse as LGBTQ+ youth, youth of color, and young men. The prevalence of despair in youth is similarly high, with significant disparities by race, sex, and gender identity. In 2024, the CDC reported that 40% of high school students reported persistent feelings of sadness or hopelessness during the past year. This was higher in females (53%) than males (28%); higher among LGBTQ+ (65%) than cisgender and heterosexual (31%) individuals; highest among American Indian or Alaskan Native (45%), Hispanic (42%), multiracial (41%), Black (40%), White (39%), and Asian (32%) individuals; and lowest among Native Hawaiian or Pacific Islanders (26%) [3,5]. 

We share our perspective as psychiatrists practicing in San Francisco, long hailed as a city championing social activism and compassionate ethos, where progress made over the decades is being toppled at an alarming rate. The current Trump administration has ushered in what might be considered a post-DEI era, enacting numerous executive orders, leading to the dismantling of diversity, equity, and inclusion (DEI) efforts, restrictions on gender affirming care, reversal of transgender policies, deportation and detention of immigrants, and rollback of research funding and health insurance coverage [6,7,8]. We know that self-harm rates are worse in marginalized groups such as youth of color and LGBTQ+ adolescents, but we are losing funding to both study and treat these populations, as a loud and powerful majority center White men as perceived victims [9,10]. As resentment against marginalized groups mounts and the narrative of victimhood becomes increasingly universalized, we explore why different groups, regardless of their identity, might see themselves as vulnerable and ostracized. We highlight how groups of youth as disparate as youth of color, LGBTQ+ youth, and young men experience unique sociopolitical stressors that contribute to increased suicidality. We argue that despite differences in their contexts, they experience shared pathways to suicide. Of note, we do not discuss intersectionality in the scope of our paper but recognize this is an important consideration that must be explored. In response to regressive policies that strip away resources from marginalized groups, we propose that mental health professionals must not only advocate for protective policies but also collaborate with the education and public health sectors to strengthen programs such as the Social Emotional Learning curriculum, which is foundational in building a nurturing school climate.

## 2. Theoretical Frameworks for Understanding Suicide

There have been dramatic shifts over the years in how society has viewed suicide, from the older religious perspective that taking one’s own life is a sin to the current day’s strong influence of the psychological model, which purports that mental illness is the primary cause, the so-called “90 percent statistic” [11]. In ancient Rome, some forms of self-killing were considered honorable, with the Epicureans acknowledging that one had the right to dispose of their own life. Laws permitted self-killing for a variety of reasons, including *taedium vitae* (one has had enough of life), *inpatientia* (bodily suffering), *dolor* (psychic pain), and *pudor* (shame) [12,13]. In 1897, Durkheim proposed that the root cause of suicide might be underlying social conditions, which heralded the move away from the religious belief that condemned it to be a moral crime. He proposed four different types of suicide based on the lack or abundance of social integration and moral regulation: egoistic suicide, altruistic suicide, anomic suicide, and fatalistic suicide [13,14,15,16]. Joiner talks about “thwarted belonging,” also in the context of isolation and non-integration, as a cause for suicide [17]. While it is considered in suicidology to be a well-established fact that 90% of all suicides are a consequence of mental disorders, Hjelmeland and Knizek argue that, in addition to psychiatric conditions, one has to focus on the contextual and the relational in the life course perspective to fully understand the nature of suicide [18]. There have been many theories to better understand suicidal thoughts and behaviors, but predictive ability has not improved despite a half-century of risk factor research [19]. There is no simple solution to decrease suicidality and increase mental well-being across a wide range of youth groups, yet ongoing research and theories are critical to refining and developing intervention and prevention efforts. A cohesive way to approach how diverse groups experience mental health challenges is to understand the social determinants of mental health, an approach that is aligned with Durkheim and Joiner’s views that the sociopolitical context is inextricably tied to suicide.

## 3. Sociopolitical Forces Eroding Belonging in Disparate Youth Groups

Cover elegantly puts together a narrative for rethinking minority youth suicide, drawing upon themes of loneliness, isolation from the works of Durkheim and Joiner, and Butler’s approach to social belonging, livability, and grievability [20]. Butler defines livability as sociopolitical, economic, institutional, biological, and psychological conditions that enable a life of meaning rather than mere existence, while unlivability is characterized by a loss of these vital conditions, leading to extreme suffering and injustice. When one is faced with an unlivable life, life may not feel worth safeguarding, protecting, or valuing, hence not worth grieving or “ungrievable” [21]. Rather than viewing isolation and loneliness as the root causes of suicide, Cover articulates that they may be reconceptualized as contributions to unlivability [20]. Together, these frameworks suggest that disparate populations may face suicide risk via common pathways toward an “unlivable life,” which may be paved by sociopolitical conditions that erode belonging, deprive resources for basic needs, or impose persecution.

### 3.1. Youth of Color

The suicide rate among American Black youth between 10 to 17 years old increased by 144% between 2007 and 2020, significant for being the fastest-growing rate among racial groups [22]. Joseph and colleagues showed that from 1999 to 2020, suicide rates among Black females aged 15 to 84 increased from 2.1 to 3.4 per 100,000, with a concentrated rate increase from 1.9 to 4.9 per 100,000 among those aged 15 to 24. They utilized age, period, and cohort (APC) effects to elicit key information, which helped with more precise identification of the concentration of suicide risk for young Black women in the US [23].

The sociopolitical context has had a marked impact on suicide rates among youth of color in the United States. Structural factors such as racism, concentrated poverty, and economic segregation are strongly associated with suicide risk [24,25]. The CDC’s Youth Risk Behaviors Survey from 2023 found that students who experienced racism had a higher prevalence of indicators of poor mental health, substance use, and suicide risk. Among students of color, including American Indian or Alaska Native, Asian, Black, Hispanic, and multiracial students, those who experienced racism had two times higher prevalence of seriously considering and attempting suicide compared with those who never experienced racism [26,27]. Moreover, this group’s disparities in access to healthcare and education may be further exacerbated by disproportionate representation of youth of color in the foster care and juvenile incarceration systems, settings known to be associated with higher suicide risk [28,29]. Beyond systematic forms of racism, many American youth of color experience insidious forms of discrimination in daily life, such as through online racial discrimination on social media; these forms of racism are also associated with increased suicidal ideation among Black and Latinx youth [30,31,32].

Recent sweeping policy changes from the current administration have compounded an already vulnerable social foundation. Executive orders terminating diversity, equity, and inclusion (DEI) programs and initiatives have unilaterally retracted critical programs that ameliorated inequities for minoritized youth and halted research documenting race- and ethnicity-disaggregated research on mental health outcomes for particular minoritized groups [33]. Moreover, the administration’s targeting of news outlets, suppression of dissent, threats of funding cuts to academic institutions for DEI efforts, and aggressive enforcement of anti-immigration policies have exerted a “chilling effect” on minoritized Americans [34,35,36,37]. These executive orders codify the erasure of youth of color from public and academic awareness, withdraw resources, and mechanize discrimination, all factors that contribute to suffering, injustice, and unlivability.

### 3.2. LGBTQ+ Youth

Mental health challenges and suicidality are significant concerns for LGBTQ+ youth, a population that has its own unique history and sociopolitical context. It is important to understand the underlying role stigma and minority stress play in determining mental health disparities among LGBTQ+ youth. Meyer posited the minority stress theory in 2003, by which individuals from stigmatized social categories are exposed to excess stress as a result of their social and often minoritized position [38,39]. These unique stressors go beyond the general stressors that are experienced by all individuals in that they are both socially based and chronic throughout one’s lifetime, given their perpetuation through laws and policies. Therefore, any discussion of mental health challenges present in LGBTQ+ youth must acknowledge that the presence of these challenges is significantly impacted by targeted stigma operating on an individual, interpersonal, and structural basis.

On an individual level, internalized homophobia and transphobia, which refer to the internalization of negative societal attitudes about one’s sexual orientation and gender identity, can lead to unhealthy behaviors and poor mental health outcomes among LGBTQ+ individuals. For example, Perez-Brumer et al. show internalized transphobia is associated with increased risk of lifetime suicide attempts among transgender adults [39]. Another example of the impact of stigma on the individual level is rejection sensitivity. As a consequence of previous experiences with prejudice and discrimination toward their LGBTQ+ group membership, many individuals learn to anxiously anticipate rejection and develop unhealthy coping strategies as a result [40].

Perhaps one of the most salient examples of interpersonal stigma leading to poor mental health outcomes in LGBTQ+ youth is bullying and victimization. To draw from data reflecting another Western country, in British LGBTQ+ youth, bullying and online bullying have been shown to be correlated with self-harm and suicidal ideation [41]. In 2024, the Trevor Project reported that nearly half of LGBTQ+ teenagers in the US experienced bullying and demonstrated higher rates of suicide attempts than their counterparts who did not experience bullying [42]. Jadva et al. found that nearly 17% of UK adolescents who experienced more homophobic, biphobic, and transphobic (HBT) bullying were more likely to report self-harm and suicidal ideation, while 20% were more likely to report a suicide attempt. These results are even stronger with regard to online bullying, with adolescents who experienced online bullying being 25% more likely to report self-harm and suicide ideation and 22% more likely to report a suicide attempt [41].

The Trump administration ushered in America’s post-DEI era, enacting numerous executive orders rolling back DEI initiatives and suppressing recognition of gender and sexual minorities, amplifying negative structural effects on LGBTQ+ mental health. Recent cuts in federal funding for public health care have jeopardized targeted resources for minoritized populations, such as the discontinuation of the LGBTQ+ youth suicide hotline on 17 July 2025 and limitations on gender affirming care. Given limited data on these very recent changes on federal, state, county, and city levels, only an anecdotal discussion based on clinical and academic experiences can be shared. As recently as Spring 2025, the academic clinical staffing within our city’s LGBTQ+ clinic serving publicly-insured patients was severely cut as a result of shifting national and state funding contracts. As a result of these political shifts, both public and private clinics serving LGBTQ+ identifying patients across the California Bay Area have seen exacerbations in mental health symptoms, from depression and anxiety to trauma and gender dysphoria. These symptom exacerbations have been experienced in waves that correlate with social and political discourse centered on sexual and gender identity and threats to their legal protections. Some states like California are stepping up to fill the gap, but without federal support, our nation will face a patchwork quilt of mental health care policy deserts for minoritized youth [43]. As such, discriminatory policies systematize structural stigma and inevitably undermine public mental health [44].

### 3.3. Young Men and Boys

The sociopolitical context of young men and boys has shifted drastically in the last several decades. Economically, technological advances in job automation have decreased well-paying manufacturing and manual labor work that were predominantly occupied by men; these trends are likely to be ongoing with the spread of generative artificial intelligence technology. The COVID-19 epidemic additionally exacerbated economic pressures, diminished social ties, and negatively impacted civic and social communities. These changes intersect with gendered norms such as R.W. Connell’s conception of hegemonic masculinity, which emphasizes stoicism, toughness, and control [45]. These norms socialize boys and men to suppress emotional vulnerability, which in turn reduces help-seeking behaviors for depression, anxiety, and suicidal ideation [46]. Young men find themselves in the “double bind” of masculinity, where conforming to masculine norms leaves them bereft of care, but eschewing these norms alienates them from their peers [47]. These shifts have led to significant declines in mental health outcomes for young men.

In 2023, the suicide rate among American men was nearly four times higher than women, accounting for 80% of all suicides [48]. Explanations for the suicide rate disparity between men and women have been hypothesized to be related to the higher lethality methods that men are more likely to employ (e.g., hanging, asphyxiation, and by firearm). Other possible reasons for high-lethality behavior in males may be: a stronger intent to die, a greater threshold for help-seeking, social isolation with a lower likelihood of being rescued, more aggressive personality traits, the involvement of alcohol, and the significant role of unemployment [49]. Young men are increasingly facing disconnection from educational, employment, civil, and social life. Compared to women, men in the US face higher unemployment rates and have seen a precipitous drop in college enrollment and completion over the last decade, with over 10% of men between the age of 25 and 54 neither working nor in school. Unemployment, lack of a college education, and financial hardship are all respectively associated with triple, quadruple, and quintuple the odds of suicide attempts [50,51,52,53,54]. Data from a population-wide study in Germany showed that nearly a quarter of men report social isolation, both through objective metrics such as living alone and being unmarried and subjective reports of loneliness [55]. These compounding factors of unemployment, reduced education, and increased loneliness drive the majority of suicides in young men [55,56].

This erosion of social and civic connection coincides with a broader shift toward life online, where American youth aged 8 to 18 now spend an average of 7.5 h per day on digital platforms [57]. For many young men, loneliness and disconnection increase susceptibility to online radicalization. Research links vulnerability to extremism with lower levels of social capital, and extremist online communities show a high concentration of mental health difficulties [58]. The involuntary celibate (“incel”) community offers a stark example.

The term “involuntary celibate” was originally coined by a Canadian female undergraduate student in 1997. It was then adopted by a group of heterosexual men who coalesced around their frustration from romantic rejection and pointed to several perceived causes of this rejection, which ranged from their own physical appearances to larger societal movements such as feminism. This online subculture first emerged on Reddit, with 40,000 active members before it was banned in 2017; online membership then moved to numerous other platforms such as 4chan, 8chan, and privately encrypted platforms such as Telegram and Discord. Since then, the incel community has radicalized significantly, adopting extremist ideologies including misogyny, racism, and in rare cases, terrorism. Emerging research demonstrates that factors such as social alienation, perceived lack of social mobility, algorithm-driven dating technology, and online echo chambers have heightened mating market competition and engendered hopelessness, both of which are associated with increased suicidal ideation [59,60,61,62,63].

Increasingly, studies point to a significant burden of mental health challenges in the incel community. Despite public perception, most incels do not engage in violence. Exploratory studies show incels report a high prevalence of being bullied, greater rejection sensitivity, a greater fear of being single, and lower levels of self-esteem and secure attachment than their male counterparts. Incels’ self-reported prevalence of depression and anxiety is as high as 95% and 94%, respectively; in comparison, a representative sample of American adults reported a prevalence of depression and anxiety of 28% and 35% to the CDC in 2020 [64,65]. Only about half of incels have reported engaging in psychotherapy but are ten times less likely to report benefit from it compared to the majority of American adults who have engaged in psychotherapy [60,64,65,66]. 

The incel community exhibits a complex relationship with suicidality, in which high baseline vulnerability is compounded by online discourse that both offers support and normalizes self-harm. Within incel forums, an analysis of 80 suicide posts by incels, Daly and Laskovstov found that incel communities were complex in their dual role as a support for lonely and hopeless incels and also sometimes as pro-suicide spaces where “roping” was encouraged [67]. While this particular community may find themselves holding demographic markers with higher social capital (cis-gendered men, heterosexual, predominantly Caucasian), the discussion above demonstrates how loneliness and hopelessness may drive their radicalization and subsequent poor mental health outcomes.

## 4. Shared Pathways Toward Suicidality

Although the contexts of youth groups as disparate as youth of color, LGBTQ+ youth, and young men feature distinct histories and sociopolitical pressures, their pathways toward suicidality converge along several shared mechanisms. When one considers Joiner’s ideas of “thwarted belonging,” these groups of youth demonstrate social alienation through the influences of racism, homophobia, bullying, and in the disconnection from social communities and meaningful relationships. In an effort to better understand the increase in self-harm tendencies in sexual minorities, the psychological factors such as self-esteem and thwarted belongingness have been shown to have a greater association, and may be helpful in directing preventative and interventional efforts [68,69]. Butler’s description of the “precariousness” of life and argument that sustaining a “livable” life necessitates affirming socio-economic and political conditions is evident through the transformative and injurious effects of the COVID-19 epidemic and recent systematic national policies ossifying race-, sex-, and gender-based discrimination. In 2022, Case and Deaton echoed Butler’s ideas when they coined the term “deaths of despair” to describe the connection between the deterioration of life for many White male Americans and rising deaths from substance use overdoses and suicide in the latter part of the 20th century [70]. Their selective and narrow depiction of these deaths as disproportionately affecting White men led Trump’s first-term administration to focus on opioid use disorder treatment efforts, but in ways that predominantly served White Americans [71]. There have been many reports that have refuted this data, including Muennig and colleagues, who offer a more expansive and enduring narrative of the opioid crisis that highlights deaths of despair are “neither a recent problem nor is it confined to Whites.” Their work identifies disturbing trends in declining relative life expectancy in the United States and is part of a large body of literature connecting this excess mortality from despair to suicidality [72].

The ideas of these thinkers and researchers highlight the inherent interconnectedness of our lives to each other and to the sociopolitical environments we live in. While it may be unconventional to draw parallels between these disparate groups of youth, their shared end outcomes of increased rates of self-harm and suicidality, in part brought about by isolation and loneliness, exemplify Butler and Cover’s “unlivability” of life.

## 5. Recommendations and Next Steps

The intersubjective conditions of a livable life in the world imply a social and ethical obligation toward the life of the other [21]. When the social fabric supporting those with limited resources is stripped, and a government no longer honors obligations to those who need it the most, advocacy work to nurture the conditions that sustain life becomes even more vital.

Amelioration of upstream factors like poverty, for example, is correlated with declines in suicide risk, highlighting the necessity of addressing the sociopolitical context in efforts to reduce suicide risk in youth [24]. In the current American sociopolitical landscape, it is also important to recognize how policy and funding changes will limit our ability to formally study the mental health outcomes in youth of color and LGBTQ+ youth and sustain dedicated clinical spaces for their treatment. Mental health professionals can act by advocating for reforms in public policies and for changes in social norms. Shim and Compton posed the question, “If not now, when? If not us, who?” in 2020 with an urgency that is even more relevant today [73].

### 5.1. Policy Advocacy

Mental health clinicians are in a unique position to advocate for a myriad of policies that support public mental health. As administrative executive orders terminate DEI and gender equity and sexual identity inclusiveness initiatives across industry, schools, and healthcare, essentially codifying structural discrimination, it becomes increasingly vital that mental health clinicians engage in political advocacy work against such oppressive policies. The links between discriminatory policies and worse mental health outcomes cannot be ignored. For example, English and colleagues conducted an analysis showing states with more indicators of structural racism (e.g., residential segregation, incarceration rates, and educational attainment) and anti-LGBTQ+ policies (e.g., HIV criminalization, conversion therapy, and permitting hate crimes) are associated with suicide risk factors such as depressive symptoms, perceived burdensomeness, thwarted belongingness, and suicide attempts in young Black sexual minority men [74].

Just as one might prescribe therapy or pharmacotherapy to treat a mood disorder, interventions such as anti-discrimination policies have been shown to protect mental health outcomes [44]. For example, state policies that protect gender-diverse people have been associated with decreased discrimination against and reduced rates of suicidality in gender-minority individuals living in those states [75,76]. Similarly, interventions to improve the mental health of youth of color in the United States must address the underpinnings of racism within healthcare, disparities in access to quality mental health care, housing and education inequities, and the disproportionate funneling of youth of color into the U.S. incarceration system [77]. For example, investments in interventions such as multisystemic therapy, an intensive community- and family-based therapy modality which addresses risk factors across family, peer, school, and community contexts, have demonstrated reductions in recidivism and mental health issues, and improved functioning for juvenile justice involved youth [78,79,80].

The high suicide rates of young men also necessitate policies that target the reconnection of this large group of youth to schools, work, and relationships. Several states have begun to recognize this. For example, this year, the Governor of California signed an executive order addressing the “growing crisis of connection and opportunity for men and boys” [81]. Such orders are a first step to generate pathways for youth to enter education, civic life, and the workforce, expand access to mental health care, and recruit positive role models into positions of mentorship for young men.

### 5.2. Education

Considering the above discussion on individual, interpersonal, and structural stigma and their role in precipitating poor mental health outcomes, we advocate for interventions centered within the education sector that could target all three levels. Given the existing, large-scale implementation of universal school-based social emotional learning (USB SEL), mental health clinicians have a unique opportunity for collaboration to better tailor these curricula to target outcomes such as self-harm and suicidal ideation present across the youth groups discussed above. USB SEL has the potential for large-scale impact not only due to its universal implementation across many schools around the country but also because of its ability to teach youth core concepts within psychoeducation.

Fricker introduced the concept of epistemic injustice, which included testimonial injustice (unjust deficit of credibility owing to prejudice) and hermeneutical injustice (one’s inability to give meaning and/or communicate their experience because their marginalized social group lacks the collective conceptual resources to interpret it) [82]. When one considers the idea of epistemic uncertainty—that a lack of knowledge about a situation or system gives rise to uncertainty within an individual and limits their ability to verbalize their experience—it is clear that helping youth understand their internal mental health experiences could increase their ability to access care and reduce their isolation. Encouraging skill building in this arena from an early age through school is a powerful way to encourage youth to develop the self-knowledge and communication skills for a livable and connected life [83].

USB SEL interventions are designed to support the development of intrapersonal and interpersonal skills to promote psychological health for all students in a given school or grade. Cipriano and colleagues’ meta-analysis of over 400 studies of USB SEL found that students who participated in USB SEL interventions experienced statistical improvement in school climate and safety, civic attitudes and behaviors, SEL skills, peer relationships, attitudes and beliefs, prosocial behaviors, externalizing behaviors, emotional distress, school functioning, and academic achievement [84]. These outcomes are particularly bolstering for the role of USB SEL when one considers UK-based data that adolescents who reported a more positive school experience were 40% less likely to report self-harm or suicide attempts and 65% less likely to report suicidal ideation [41]. Across disparate groups of youth, further development and improvement of peer relationships, emotional distress, and prosocial behaviors could benefit their mental health outcomes.

One example of potential collaboration could be the incorporation of the findings from Brennan et al.’s systematic review and thematic meta-synthesis into the appropriate USB SEL skill teachings [85]. Brennan and colleagues identified the importance of breaking the link between a person’s current psychological state and the act of self-harm as a key point of intervention in reducing self-harm. Relevant skills such as shifting focus, substituting physical actions, and managing provocations could be taught under the competency areas identified by the Collaborative for Academic, Social, and Emotional Learning (CASEL) framework: self-awareness, self-management, and responsible decision making [84,85,86]. A second example may be to consider Schacter and colleagues’ ways of enhancing episodic memory, a life-oriented thought pattern that has been studied in suicidal individuals, which describes the ability to imagine personal events in both the past and future [87]. In an increasingly turbulent world, it is difficult to imagine a better tomorrow, let alone plan for it, especially for the young. For this reason, it is critical that USB-SEL programs help youth incorporate enhancements to episodic memory by providing brief training in recollecting details of past experiences and bridging them to creative thinking and problem-solving for the future. These skills are the foundational elements to instill hope and imagine a more livable future.

## 6. Discussion and Conclusions

Our perspective makes a case for shared pathways to suicidality across disparate youth groups, drawing on traditional suicide paradigms. The scope of our paper is limited to the United States, with some studies drawing on research from mainly Westernized countries. In the future, it may be illuminating to explore parallel policy developments and their outcomes for minoritized youth groups in other countries, within the context of their unique sociocultural–political settings. We also acknowledge the limitation posed by this paper’s lack of a first-person lived experience perspective in the form of a case or vignette that could highlight the influence that the American sociopolitical landscape has had on suicidal behavior. 

We acknowledge that our perspective is informed by our strong belief in the importance of sociopolitical structures and policies that promote the freedoms of individuals with minoritized identities, including DEI efforts. Our use of the term “post-DEI” is based on our observations of the recent sociopolitical shifts of the current Trump administration toward dismantling numerous DEI policies. While exploring alternative views is outside the scope of our perspective piece, we recognize many may hold different beliefs regarding DEI policies, including views that “merit, excellence, and intelligence” (MEI) should be prioritized in hiring and admission decisions, DEI efforts violate Constitutional protections, and DEI policies could generate backlash [88,89,90].

Current sociopolitical shifts have drastically destabilized what was once foundational support for vulnerable groups of youth, sowing fear and uncertainty. These recent shifts have further disintegrated social bonds and structures, promoting an increasingly atomized society. We have sought to highlight how lost meaningful connections to self, others, and society carve a shared pathway towards suicidality across very disparate youth groups; groups who are considered outliers, whether perceived by themselves or others, are particularly vulnerable. The intersubjective nature of our shared lives as humans allows us to see and verbalize what would be an unlivable life for any of us. As mental health professionals, we must lean into advocacy to build livable conditions of life within our sociocultural context and equip youth with the skills to understand and articulate their internal and external experiences, build relationships, seek support, and navigate an ever-changing and precarious landscape.

## Data Availability

No new data were created or analyzed in this study.

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
