# Peer review of "Is Life Unlivable for Youth in Post-DEI America?: Understanding Rising Suicide Rates Across Diverse Youth Groups Through Traditional Suicide Paradigms"

_healthcare, 2025, doi:10.3390/healthcare13202585_

Round 1
Reviewer 1 Report
Comments and Suggestions for Authors
Overall, this is a timely article that addresses a critical, socially relevant public health issue. I think the content is well-organized and is structured in a way that addresses the authors' main points in a compelling narrative.
Mata et al submitted a Perspective titled: "Is Life Unlivable for Youth in Post-DEI America?: Understanding Rising Suicide Rates Across Diverse Youth Groups 3 Through Traditional Suicide Paradigms." This article sought to address and discuss if life has become unlivable for American youth, grounding their discussion in prominent theoretical paradigms proposed by Durkheim, Joiner, and Butler. Using these psychological models for suicide, the authors provided a well-defined perspective on common themes and new hardships that shape suicidal behaviors in various groups of Americans. I think the citations included were relevant, and accurate given the topic material. I do think there is an opportunity to discuss deaths of despair as posed by Case and Deaton (2020) and resulting articles, however this topic is adjacent to the author's point and does not necessarily warrant and in-depth analysis, though it could supplement their overall point about rising contributions to mortality in adolescents. This perspective is VERY relevant and topically important. We often focus on data-driven (numbers and statistics) related to suicide and accompanying behaviors, which is useful. However, Mata et al provided not only a theoretical basis for these changes but a cultural context that outlines why these behaviors may be occurring across cultural, gender, and other adolescent groups, which is comprehensive, yet focused and working toward answering their central question about livability. This perspective adds to other published literature discussing current, relevant topics in the field of youth suicide and associated behaviors, including touching on mental health and help-seeking behavior. It provides context for important discussions related to suicide, help-seeking behavior, and the greater mental health context, providing solutions for addressing prevention (i.e., universal school-based social emotional learning). This particular article was a perspective, which established a thorough review of literature and psychological models to support their conclusions. While not an original research article, I think the application of current perspectives in psychology and evidence from recently and historically published literature. Additionally, there were a few points in the article where my initial review suggested they include a reference for clarity and to strengthen where the supporting evidence is coming from. I do feel the authors should address those points. The article comes to an appropriate conclusion that summarizes the authors ' experiences, review of current literature and poses a brief, succinct push to lean in to advocacy to continue to build livable conditions for America's youth. This article brilliantly pulls in current social contexts, politically relevant policy, and interweaves their conclusions with appropriately supporting field-based theoretical themes. Overall, the references are appropriate. Though my original review does suggest some points in the article where they should cite or clarify where certain statements came from. This perspective did not include any figures or tables, which I think is appropriate for an article labeled as a perspective. They could add a diagram that shows the overlap of themes from combining theoretical approaches, or highlighting cross-group commonalities; however, I do not think this will necessarily add anything groundbreaking to the article that was not already well-stated by the authors. Overall, outside of a few minor corrections to grammar and formatting, I think this article should be accepted for publication.Suggestions and Corrections:
In the introduction lines 37-39, when talking about persistent feelings of sadness or hopelessness, the percentages for Black, White, and Asian are double-listed. Please correct for consistency.
Line 48, at the plus to "LGBTQ," for consistency.
Lines 93-95 the sentence "Rather than viewing isolation and loneliness as the root causes of suicide, Cover articulates they may be reconceptualized as contributions to unlivability." should have a citation for Cover at the end of the sentence.
Did the information in the first paragraph under section 3.1 Youth of Color lines 100-107 all come from the same source, cited as (18). If not please include the specific citations for each source of statistics.
Line 199 should read "four times", add s to time.
Author Response
Reviewer 11. Overall, this is a timely article that addresses a critical, socially relevant public health issue. I
think the content is well-organized and is structured in a way that addresses the authors' main
points in a compelling narrative.
Mata et al submitted a Perspective titled: "Is Life Unlivable for Youth in Post-DEI America?:
Understanding Rising Suicide Rates Across Diverse Youth Groups 3 Through Traditional Suicide
Paradigms." This article sought to address and discuss if life has become unlivable for American
youth, grounding their discussion in prominent theoretical paradigms proposed by Durkheim,
Joiner, and Butler. Using these psychological models for suicide, the authors provided a
well-defined perspective on common themes and new hardships that shape suicidal behaviors in
various groups of Americans. I think the citations included were relevant, and accurate given the
topic material.
Author response: Thank you!
2. I do think there is an opportunity to discuss deaths of despair as posed by Case and Deaton (2020)
and resulting articles, however this topic is adjacent to the author's point and does not necessarily
warrant and in-depth analysis, though it could supplement their overall point about rising
contributions to mortality in adolescents.
Author response: We agree discussing deaths of despair is a key adjacent issue that provides a
helpful supplementation to our overall point about rising contributions to mortality in adolescents.
We have integrated a brief discussion about Case and Deaton’s ideas, in addition to those of
Muenning et al’s in expanding our understanding of these deaths of despair to include a broader
group of people. [Line 320-333]
3. This perspective is VERY relevant and topically important. We often focus on data-driven
(numbers and statistics) related to suicide and accompanying behaviors, which is useful.
However, Mata et al provided not only a theoretical basis for these changes but a cultural context
that outlines why these behaviors may be occurring across cultural, gender, and other adolescent
groups, which is comprehensive, yet focused and working toward answering their central
question about livability. This perspective adds to other published literature discussing current,
relevant topics in the field of youth suicide and associated behaviors, including touching on
mental health and help-seeking behavior. It provides context for important discussions related to
suicide, help-seeking behavior, and the greater mental health context, providing solutions for
addressing prevention (i.e., universal school-based social emotional learning). This particular
article was a perspective, which established a thorough review of literature and psychological
models to support their conclusions. While not an original research article, I think the application
of current perspectives in psychology and evidence from recently and historically published
literature. Additionally, there were a few points in the article where my initial review suggested
they include a reference for clarity and to strengthen where the supporting evidence is coming
from. I do feel the authors should address those points. The article comes to an appropriate conclusion that summarizes the authors ' experiences, review of current literature and poses a
brief, succinct push to lean in to advocacy to continue to build livable conditions for America's
youth. This article brilliantly pulls in current social contexts, politically relevant policy, and
interweaves their conclusions with appropriately supporting field-based theoretical themes.
Overall, the references are appropriate. Though my original review does suggest some points in
the article where they should cite or clarify where certain statements came from. This
perspective did not include any figures or tables, which I think is appropriate for an article labeled
as a perspective. They could add a diagram that shows the overlap of themes from combining
theoretical approaches, or highlighting cross-group commonalities; however, I do not think this
will necessarily add anything groundbreaking to the article that was not already well-stated by the
authors. Overall, outside of a few minor corrections to grammar and formatting, I think this
article should be accepted for publication.
Author response: We appreciate the reviewer’s thoughts regarding the scope and content of this
perspective piece. We have highlighted grammatical edits to our paper below and agree that a
diagram could be helpful, though not necessary, to illustrate the overlap of themes from the
theoretical perspectives highlighted. We have elected not to include a figure and appreciate the
reviewer’s remarks on the clarity of these themes in the written piece.
4. Suggestions and Corrections:
In the introduction lines 37-39, when talking about persistent feelings of sadness or hopelessness,
the percentages for Black, White, and Asian are double-listed. Please correct for consistency.
Author response: We have corrected the double-listed percentages for persistent feelings of sadness
or hopelessness in Black, White, and Asian individuals to ensure consistency [Line 41-42].
5. Line 48, at the plus to "LGBTQ," for consistency.
Author response: We have corrected this discrepancy and ensured “LGBTQ+” is used throughout
for consistency [Lines 34, 53]
6. Lines 93-95 the sentence "Rather than viewing isolation and loneliness as the root causes of
suicide, Cover articulates they may be reconceptualized as contributions to unlivability." should
have a citation for Cover at the end of the sentence.
Author response: Thank you for this catch. We have added a citation for Cover at the end of this
sentence [Line 111].
7. Did the information in the first paragraph under section 3.1 Youth of Color lines 100-107 all
come from the same source, cited as (18). If not please include the specific citations for each
source of statistics.
Author response: We have added a citation for the first sentence “The suicide rate among Black
youth between 10 to 17 years old increased by 144% between 2007 and 2020, the fastest-growing rate among racial groups” [Lines 117-119]. The statistics for the rest of the paragraph are all pulled
from Joseph et al, which is cited at the end of the paragraph [Line 125].
8. Line 199 should read "four times", add s to time.
Author response: We have edited “four time” to “four times” [Line 237].
Reviewer 2 Report
Comments and Suggestions for Authors
The authors have prepared a very well-written perspective on the potential impacts of current political environment on the well-being of vulnerable or diverse youth groups, and strategies that could be implemented to reduce adverse policy impacts. The authors review well the scientific evidence supporting universal school-based social and emotional learning, though this reviewer wonders why this intervention is not mentioned in the title?
In terms of reporting, it cannot be stated that state policies "decreased" discrimination (page 7, line 295) since the citations used to support this statement can only suggest an association. References 61 and 62 are not controlled evaluations.
Two minor edits for you to consider are:
page 4, line 186: The sociopolitical context HAS shifted, not have (not plural since the noun/subject is singular).
page 8, line363, I wonder if 'foundational' instead of foundation would be better?
Author Response
Reviewer 2
1. The authors have prepared a very well-written perspective on the potential impacts of current
political environment on the well-being of vulnerable or diverse youth groups, and strategies that
could be implemented to reduce adverse policy impacts. The authors review well the scientific
evidence supporting universal school-based social and emotional learning, though this reviewer
wonders why this intervention is not mentioned in the title?
Author response: We appreciate this review. We have decided not to focus the title of this paper on
the interventions we have highlighted, as they are a few of many possible responses to the trends
toward increased suicidality in adolescents. For example, we also make a case for the importance of
policy advocacy alongside interventions in education. We aim to focus our paper and title on how
traditional suicide paradigms can provide a unifying model for understanding rising suicidality
across diverse groups of youth. We hope that revisiting these paradigms, which emphasize the
sociopolitical influences on suicidality, may allow the reader to imagine different interventions that
can aid our youth in leading more livable lives - while universal school-based social and emotional
learning is one of these promising interventions, it is not the only path, and thus we have decided
not to focus our title on this intervention.
2. In terms of reporting, it cannot be stated that state policies "decreased" discrimination (page 7,
line 295) since the citations used to support this statement can only suggest an association.
References 61 and 62 are not controlled evaluations.
Author response: We have added “have been associated with” to reflect association but not
causation in this statement [Line 374]. The sentence now states “For example, state policies that
protect gender-diverse people have been associated with decreased discrimination against and
reduced rates of suicidality in gender-minority individuals living in those states.”
3. Two minor edits for you to consider are:
page 4, line 186: The sociopolitical context HAS shifted, not have (not plural since the
noun/subject is singular).
Author response: We have edited “have” to “has.” [Line 221]
4. page 8, line363, I wonder if 'foundational' instead of foundation would be better?
Author response: We have edited “foundation” to “foundational” [line 476].
Reviewer 3 Report
Comments and Suggestions for Authors
The manuscript is a timely, current perspective that would contribute literature. It highlights connections between policy changes and mental health, with a strong emphasis on suicide. I provide several suggestions to improve it.
- It would be helpful if you add study findings on effects of antiDEI policy on suicide rates minorities along with the opinion and perspective papers. If no such study is yet present, it should be added as limitations.
- As this is a perspective paper, author should also acknowledge their potential bias in interpreting "post-DEI" era , and they should cite contrary opinions (e.g., policy proponents' views) for making the paper more balanced.
- Be consistent in using LQBT terms throughout paper.
- The paper may also consider the problems encountered in other countries that have parallel policy developments and their outcome with respect to minorities’ suicidal approaches, quality of life and livability.
Author Response
Reviewer 3
1. It would be helpful if you add study findings on effects of antiDEI policy on suicide rates
minorities along with the opinion and perspective papers. If no such study is yet present, it should
be added as limitations.
Author response: We appreciate this critical point and agree that citing existing research on the
detrimental effects of anti-DEI policies could further strengthen and contextualize our arguments
linking discriminatory policies to suicide rates. We have highlighted one such paper examining the
impact of structural racism and anti-LGBTQ+ policies on suicide rates in young Black sexual
minority men in our section on Policy Advocacy. The addition reads as follows:
“The links between discriminatory policies and worse mental health outcomes cannot be
ignored. For example, English and colleagues conducted an analysis showing states with
more indicators of structural racism (eg. residential segregation, incarceration rates,
educational attainment) and anti-LGBTQ+ policies (eg. HIV criminalization, conversion
therapy, permitting hate crimes) are associated with suicide risk factors such as depressive
symptoms, perceived burdensomeness, thwarted belongingness, and suicide attempts in
young Black sexual minority men .” [Lines 363-370]
2. As this is a perspective paper, author should also acknowledge their potential bias in interpreting
"post-DEI" era , and they should cite contrary opinions (e.g., policy proponents' views) for
making the paper more balanced.
Author response: We very much agree that it is important to recognize contrary opinions,
particularly for polarized topics such as DEI policies. We have created a Discussion section to
acknowledge some of the limitations of our paper. While exploring contrary opinions is outside the
scope of our perspective, we have included a sentence to acknowledge some alternate opinions and
stated more explicitly that our use of the term “post-DEI” and our opinions are merely a
perspective:
“We acknowledge that our perspective is informed by our strong belief in the importance of
sociopolitical structures and policies that promote the freedoms of individuals with
minoritized identities, including DEI efforts. Our use of the term “post-DEI” is based on
our observations of the recent sociopolitical shifts of the current Trump administration
toward dismantling numerous DEI policies. While exploring alternative views is outside the
scope of our perspective piece, we recognize the importance of understanding why many
may hold different beliefs regarding DEI policies, including views that “merit, excellence,
and intelligence” (MEI) should be prioritized in hiring and admission decisions, DEI efforts
violate Constitutional protections, and DEI policies could generate backlash.” [Lines
464-474]
3. Be consistent in using LQBT terms throughout paper.
Author response: We have ensured “LGBTQ+” is used throughout for consistency [Lines 34, 53]
16. The paper may also consider the problems encountered in other countries that have parallel policy
developments and their outcome with respect to minorities’ suicidal approaches, quality of life
and livability.
Author response: This would be fascinating to explore! Given the unique sociopolitical context of
different countries, we believe this deserves its own paper. Though this is outside the scope of our
perspective, which is focused on the United States, we have included a line to acknowledge this in
our Discussion Section:
“Our perspective makes a case for shared pathways to suicidality across disparate youth
groups, drawing on traditional suicide paradigms. The scope of our paper is limited to the
United States, with some studies drawing on research from mainly Westernized countries.
In the future, it may be illuminating to explore parallel policy developments and their
outcomes for minoritized youth groups in other countries, within the context of their unique
socio-cultural-political settings.” [Lines 454-460]
Reviewer 4 Report
Comments and Suggestions for Authors
I think this is an incredibly important piece of work and as such support its publication.
Some missing citations, e.g. “Suicide deaths among 10 to 24-year-olds increased by 62% from 2007 to 2021.” and “It has risen dramatically in preteens as young as 8 years old, with an 8.2% annual increase from 2008 to 2022”. I noted this in a few places throughout so please ensure all claims are suitably supported and cited.
I realise this is a clinician perspective but would be good if possible to have someone with lived experience of suicidal behaviour also contributing to this work. Possibly it is too late now but some acknowledgement of this as a limitation would therefore be good.
“…with the assumption that nearly all suicides result from mental illness, although this is often diagnosed after the fact from psychological autopsies” -I don’t think this is quite accurate, most theoretical and clinical models see mental health diagnoses as contributing factors but not as fully explaining the occurrence of suicide. Instead a combination of psychological, social , and other variables are usually implicated.
“There have been many theories to better understand suicidal thoughts and behaviors to help guide prevention and treatment efforts, but predictive ability has not improved despite a half century of risk factor research”. This is fair but I would add that prediction of suicide is not always the aim of theory, and often its more about development of effective intervention or prevention.
I think the focus here is the US, but with some of the statistics presented this is unclear so please clarify which stats relate to the US and which are from elsewhere in the world.
The author suggests thwarted belongingness as a possible mechanism towards risk here. I think this is theoretically consistent but some comment on how this holds up empirically would be good. In our own research we found these is usually an association here but belongingess seems to often have a weaker or less consistent link to self-harm than other variables like self-esteem (e.g. https://www.tandfonline.com/doi/full/10.1080/13811118.2018.1515136; https://onlinelibrary.wiley.com/doi/10.1111/sltb.12823) . There is no need for the author’s to cite our work but I include it here as examples, as my own speculation is that self-esteem may be more important in explaining the link between marginalization and self-harm.
Author Response
Reviewer 4
1. I think this is an incredibly important piece of work and as such support its publication.
Some missing citations, e.g. “Suicide deaths among 10 to 24-year-olds increased by 62% from
2007 to 2021.” and “It has risen dramatically in preteens as young as 8 years old, with an 8.2%
annual increase from 2008 to 2022”. I noted this in a few places throughout so please ensure all
claims are suitably supported and cited.
Author response: We appreciate the reviewer bringing this to our attention. We have added
citations to identify the sources of these statistics [Line 30, 33].
2. I realise this is a clinician perspective but would be good if possible to have someone with lived
experience of suicidal behaviour also contributing to this work. Possibly it is too late now but
some acknowledgement of this as a limitation would therefore be good.
Author response: Thank you for this comment. We agree that it is now too late to incorporate a case
or vignette from our clinical work to further humanize and provide a lived experience element to
our perspective paper. We have included the following in our discussion to acknowledge this
comment and highlight the limitation it poses [Line 460-463]:
“We also acknowledge the limitation posed by this paper’s lack of a first-person lived
experience in the form of a case or vignette that could highlight the influence the American
sociopolitical landscape has has on suicidal behavior.”
3. “…with the assumption that nearly all suicides result from mental illness, although this is often
diagnosed after the fact from psychological autopsies” -I don’t think this is quite accurate, most
theoretical and clinical models see mental health diagnoses as contributing factors but not as fully
explaining the occurrence of suicide. Instead a combination of psychological, social , and other
variables are usually implicated.
Author response: We agree that the current phrasing of this statement perhaps overstates the role
of mental illness and does not acknowledge the other factors you have highlighted. We have
changed the language to the following [Lines 86-90]:
“While it is considered in suicidology to be a well-established fact that 90% of all suicides
are a consequence of mental disorders, Hjelmeland and Knizek argue that in addition to psychiatric
conditions, one has to focus on the contextual and the relational in the life course perspective to
fully understand the nature of suicide.”
4. “There have been many theories to better understand suicidal thoughts and behaviors to help
guide prevention and treatment efforts, but predictive ability has not improved despite a half
century of risk factor research”. This is fair but I would add that prediction of suicide is not
always the aim of theory, and often its more about development of effective intervention or
prevention.
Author response: We appreciate the prompting to highlight the role theory plays in intervention
and prevention. We have incorporated this comment in the following way [Lines 90-96]:
“There have been many theories to better understand suicidal thoughts and behaviors, but
predictive ability has not improved despite a half century of risk factor research. There is no simple
solution to decrease suicidality and increase mental wellbeing across a wide range of youth groups,
yet ongoing research and theories are critical to refining and developing intervention and
prevention efforts.”
5. I think the focus here is the US, but with some of the statistics presented this is unclear so please
clarify which stats relate to the US and which are from elsewhere in the world.
Author response: We have clarified our statistics with geographic details as follows:
[Line 30-33]: “Suicide deaths among 10 to 24-year-olds in America increased by 62% from
2007 to 2021; it has also risen dramatically in American preteens as young as 8 years old, with an
8.2% annual increase from 2008 to 2022”
[Line 117-119]: “The suicide rate among American Black youth between 10 to 17 years old
increased by 144% between 2007 and 2020, the fastest-growing rate among racial groups.”
[Lines 120-125]: “Joseph and colleagues showed that from 1999 to 2000, suicide rates
among Black females aged 15 to 84 increased from 2.1 to 3.4 per 100,000, with a concentrated rate
increase from 1.9 to 4.9 per 100,000 among those aged 15 to 24. They utilized age, period, and
cohort (APC) effects to elicit key information which helped with more precise identification of the
concentration of suicide risk for young Black women in the US.”
[Lines 129-135]: “The CDC’s Youth Risk Behaviors Survey from 2023 found that students
who experienced racism had a higher prevalence of indicators of poor mental health, substance use
and suicide risk. Among students of color, including American Indian or Alaska Native, Asian,
Black, Hispanic, and multiracial students, those who experienced racism had two times higher
prevalence of seriously considering and attempting suicide compared with those who never
experienced racism.”.
[Lines 139-143]: “Beyond systematic forms of racism, many American youth of color
experience insidious forms of discrimination in daily life, such as through online racial
discrimination on social media; these forms of racism are also associated with increased suicidal
ideation among Black and Latinx youth.”
[Lines 185-187]: “To draw from data reflecting another Western country, in British
LGBTQ+ youth, bullying and online bullying have been shown to be correlated to self-harm and
suicidal ideation.”
[Lines 187-190]: “In 2024, the Trevor Project reported that nearly half of LGBTQ+
teenagers in the US experienced bullying and demonstrated higher rates of suicide attempts than
their counterparts who did not experience bullying.”
[Lines 237-238]: “In 2023, the suicide rate among American men was nearly four times
higher than women, accounting for 80% of all suicides.”
[Lines 246-249]: “Compared to women, men in the US face higher unemployment rates and
have seen a precipitous drop in college enrollment and completion over the last decade, with over
10% of men between age 25 and 54 neither working nor in school.”
[Lines252-255]: “Data from a population-wide study in fellow Western country Germany
showed that nearly a quarter of men report social isolation, both through objective metrics such as
living alone and being unmarried, and subjective reports of loneliness.”
[Lines 258-260]: “This erosion of social and civic connection coincides with a broader shift
toward life online, where American youth aged 8 to 18 now spend an average of 7.5 hours per day
on digital platforms.”
[Lines 426-430]: “These outcomes are particularly bolstering for the role of USB SEL when
one considers UK-based data that adolescents who reported a more positive school experience were
40% less likely to report self-harm or suicide attempt and 65% less likely to report suicidal
ideation.”
6. The author suggests thwarted belongingness as a possible mechanism towards risk here. I think
this is theoretically consistent but some comment on how this holds up empirically would be
good. In our own research we found these is usually an association here but belongingess seems
to often have a weaker or less consistent link to self-harm than other variables like self-esteem
(e.g. https://www.tandfonline.com/doi/full/10.1080/13811118.2018.1515136;
https://onlinelibrary.wiley.com/doi/10.1111/sltb.12823) . There is no need for the author’s to cite
our work but I include it here as examples, as my own speculation is that self-esteem may be
more important in explaining the link between marginalization and self-harm.
Author response: Thank you for catching this opportunity to incorporate another facet of
psychological wellness (self-esteem) that affects risk. We have made the following addition [Lines
311-315]:
“In an effort to better understand the increase in self-harm tendencies in sexual minorities,
the psychological factors such as self-esteem and thwarted belongingness have been shown to have
greater association, and may be helpful in directing preventative and interventional efforts.”
We have additionally made note of several more edits we have made:
● Corrected “gobally” to “globally” [Line 27]
● Corrected “rollback on” to “rollback of” [Line 51]
● Added a citation for the “90 percent statistic” [Line 75]
● Corrected “on” to “of” [Line 126]
● Corrected the spelling of Schacter and added corresponding citation in the text [Line 443, 446]
● In our section on Youth of Color, we have added a line regarding the impact of the current
administrative changes on creating a “chilling effect,” contributing to a sense of “unlivability”:
“Moreover, the administration’s targeting of news outlets, suppression of dissent, threats
of funding cuts to academic institutions for DEI efforts, and aggressive enforcement of
anti-immigration policies have exerted a “chilling effect” on minoritized Americans.”
[Lines 160-153]
● We have added “Discussion and” to Section 6 to allow for a discussion regarding the scope and
limitations of our paper [Line 453]
● We have switched the order of the authors to reflect the authors’ contributions to the work [Line
4]
● Edited “Mental health challenges and suicidality is a high concern for” to “Mental health
challenges and suicidality are significant concerns for” [Line 158]
● Corrected “subreddit” to “Reddit” [Line 271]
● Changed “large” to “high [Line 283]
● Changed “events such as” to “of” [Line 318]
● Added “involved” [385]
● We have reworded several sections (highlighted in cyan)
○ [Lines 30-33]: “Suicide deaths among 10 to 24-year-olds in America increased by 62%
from 2007 to 2021; it has also risen dramatically in American preteens as young as 8
years old, with an 8.2% annual increase from 2008 to 2022.”
○ [Lines 71-77]: “There have been dramatic shifts over the years in how society has viewed
suicide, from the older religious perspective that taking one’s own life is a sin to current
day’s strong influence of the psychological model which purports that mental illness is
the primary cause, the so-called “90 percent statistic” (11). In ancient Rome, some forms
of self-killing were considered honorable, with the Epicureans acknowledging that one
had the right to dispose of their own life.”
○ [Lines 79-84]: “In 1897, Durkheim proposed that the root cause of suicide might be
underlying social conditions, which heralded the move away from the religious belief that
condemned it to be a moral crime. He proposed four different types of suicide based on
the lack or abundance of social integration and moral regulation: egoistic suicide,
altruistic suicide, anomic suicide, and fatalistic suicide.”
○ [Lines 86-90]: “While it is considered in suicidology to be a well-established fact that
90% of all suicides are a consequence of mental disorders, Hjelmeland and Knizek argue
that in addition to psychiatric conditions, one has to focus on the contextual and the
relational in the life course perspective to fully understand the nature of suicide.”
○ [Lines 118-119]: “...significant for being the fastest-growing rate among racial groups.”
○ [Lines 120-125]: “...suicide rates among Black females aged 15 to 84 increased from 2.1
to 3.4 per 100,000, with a concentrated rate increase from 1.9 to 4.9 per 100,000 among
those aged 15 to 24. They utilized age, period, and cohort (APC) effects to elicit key
information which helped with more precise identification of the concentration of suicide
risk for young Black women in the US.”
○ [Lines 139-135]: “...who experienced racism had a higher prevalence of indicators of
poor mental health, substance use and suicide risk. Among students of color, including
American Indian or Alaska Native, Asian, Black, Hispanic, and multiracial students,
those who experienced racism had two times higher prevalence of seriously considering
and attempting suicide compared with those who never experienced racism.”
○ [Lines 162-164]: “Meyer posited the minority stress theory in 2003, by which individuals
from stigmatized social categories are exposed to excess stress as a result of their social
and often minoritized position.”
○ [Lines 176-177]: “...internalized transphobia is associated with increased risk of lifetime
suicide attempts among transgender adults.”
○ [Lines 178-180]: “As a consequence of previous experiences with prejudice and
discrimination toward their LGBTQ+ group membership…”
○ [Lines 190-196]: “Jadva et al found that nearly 17% of UK adolescents who experienced
more homophobic, biphobic, and transphobic (HBT) bullying were more likely to report
self-harm and suicidal ideation, while 20% were more likely to report a suicide attempt.
These results are even stronger with regards to online bullying, with adolescents who
experienced online bullying being 25% more likely to report self-harm and suicide
ideation and 22% more likely to report a suicide attempt.”
○ [Lines 241-246]: “Other possible reasons for high lethality behavior in males may be: a
stronger intent to die, a greater threshold for help-seeking, social isolation with a lower
likelihood of being rescued, more aggressive personality traits, the involvement of
alcohol, and the significant role of unemployment.”
○ [Lines 419-421]: “...the development of intrapersonal and interpersonal skills to promote
psychological health for all students in a given school or grade.”
○ [Lines 428-430]: “...adolescents who reported a more positive school experience were
40% less likely to report self-harm or suicide attempt and 65% less likely to report
suicidal ideation.”